# Accuracy of Tumour-Associated Circulating Endothelial Cells as a Screening Biomarker for Clinically Significant Prostate Cancer

**DOI:** 10.3390/cancers11081064

**Published:** 2019-07-27

**Authors:** Sebastian Chakrit Bhakdi, Prapat Suriyaphol, Ponpan Thaicharoen, Sebastian Tobias Karl Grote, Chulaluk Komoltri, Bansithi Chaiyaprasithi, Komgrid Charnkaew

**Affiliations:** 1Department of Pathobiology, Faculty of Science, Mahidol University, Bangkok 10400, Thailand; 2X-ZELL, 133 Cecil Street, #06-02 Keck Seng Tower, Singapore 069535, Singapore; 3Department of Research and Development, Faculty of Medicine Siriraj Hospital, Mahidol University, Bangkok 10700, Thailand; 4Division of Urology, Department of Surgery, Faculty of Medicine, Siriraj Hospital, Mahidol University, Bangkok 10700, Thailand; 5Department of Pathology, Faculty of Medicine Siriraj Hospital, Mahidol University, Bangkok 10700, Thailand

**Keywords:** clinically significant prostate cancer, diagnostic accuracy, grey zone, liquid biopsy, PSA, tCEC, tumour-associated circulating endothelial cells, prostate cancer screening

## Abstract

Even though more than 350,000 men die from prostate cancer every year, broad-based screening for the disease remains a controversial topic. Guidelines demand that the only commonly accepted screening tool, prostate-specific antigen (PSA) testing, must be followed by prostate biopsy if results are elevated. Due to the procedure’s low positive predictive value (PPV), however, over 80% of biopsies are performed on healthy men or men with clinically insignificant cancer—prompting calls for new ways of vetting equivocal PSA readings prior to the procedure. Responding to the challenge, the present study investigated the diagnostic potential of tumour-associated circulating endothelial cells (tCECs), which have previously been described as a novel, blood-based biomarker for clinically significant cancers. Specifically, the objective was to determine the diagnostic accuracy of a tCEC-based blood test to detect clinically significant prostate cancer (defined as Gleason score ≥ 3 + 4) in high-risk patients. Performed in a blinded, prospective, single-centre set-up, it compared a novel tCEC index test with transrectal ultrasound-guided biopsy as a reference on a total of 170 patients and found that a tCEC add-on test will almost double the PPV of a standalone PSA test (32% vs. 17%; *p* = 0.0012), while retaining a negative predictive value above 90%.

## 1. Introduction

Despite a five-year survival rate of nearly 100%, prostate cancer (PCa) continues to be a serious threat to men’s health. In 2018 alone it caused more than 350,000 deaths worldwide, with both the United States and Europe reporting a 10% year-on-year rise in mortality [1,2].

PCa screening relies heavily on prostate-specific antigen (PSA) testing, an affordable and widely accessible solution that continues to divide the urology community due to a series of shortcomings that could put patients at risk without any clinical benefit.

One of the most frequently cited deficiencies of PSA screening is the test’s inherent “diagnostic grey zone”—a range of PSA values that do not allow for an unequivocal diagnosis due to a low positive predictive value (PPV). Often described as ranging from 4 to 10 ng/mL (with some studies expanding the range to 4–20 ng/mL), it leads to at least half of all prostate biopsies being performed on healthy men.

A second, much-debated shortcoming of PSA-driven diagnostics is the risk of exposing clinically insignificant cancers that remain asymptomatic during the patient’s lifetime, causing a significant portion of patients to face the potential complications of a prostate biopsy—most prominently bleeding, infection, and sepsis—without experiencing a difference in life expectancy or quality of life [1,3,4].

In a move to overcome these limitations, research has recently focused on finding adjunct tests to slot in between PSA and tissue biopsy. Ideally, they would improve PSA’s low PPV to rule in the disease more confidently, while also offering a high negative predictive value (NPV) to assertively rule out unnecessary tissue biopsies.

Well-published attempts to create such a tool include the prostate health index (PHI) assay and the 4Kscore Test, both relying on combinations of PSA and PSA precursors as well as clinical information. Circulating DNA and miRNA, as well as urine tests for various analytes, are reportedly also under development. To date, however, none of these alternatives has gained notable traction in clinical routine [5,6,7,8,9,10].

Multi-parametric magnetic resonance imaging (mpMRI), on the other hand, continues to gain acceptance as a potential PSA add-on and has even been incorporated into the European guidelines as a first-line investigation. The much-cited PROMIS trial from the United Kingdom, however, reported a relatively low NPV of 76% for clinically significant prostate cancer (csPCa), meaning it might be well-suited as a rule-in test positioned between PSA and biopsy, but not necessarily as a safe rule-out option. Other down sides include high cost, limited accessibility, and reliance on the Prostate Imaging-Reporting and Data System (PI-RADS), which suffers from significant inter-observer variability [11,12,13].

With that in mind, the present 170-patient study explores the potential of tumour-associated circulating endothelial cells (tCECs)—as blood-based biomarkers for csPCa that make full use of existing pathology practices and workflows—to fill the gap between PSA and biopsy with an easy-to-use and broadly scalable solution. 

Recently discovered among a high percentage of colon cancer patients, tCECs form a distinct link between primary tumour and systemic circulation [14]. Based on the well-established correlation between angiogenesis and tumour growth, we hypothesised that the presence of tCECs may be associated with the presence of PCa. We also theorised that tCECs may be prognostic, as the microvessel density in prostatectomies was found to correlate with Gleason score [15,16,17,18,19,20].

Previous studies into circulating tumour cells (CTCs) made similar assumptions, hypothesising that tCECs may serve useful as additional prognostic biomarkers in patients diagnosed with cancer. These studies relied on single-marker (CD146) positive isolation and single-marker immunostaining [21,22].

The present study is a distinct evolution from previous research on circulating cancer cells, combining a negative cell isolation approach not dependent on the expression of a single antigen with highly multiplexed immunostaining on standard laboratory slides [23,24]. In addition, the present study introduces a pre-defined immunocytopathological classification combining well-established morphological hallmarks of tumour-endothelial cells, such as nuclear aneuploidy, with well-established immune phenotypes on endothelial cells observed in the microvessels of prostate cancer [16].

To our knowledge, this is the first prospectively blinded screening study investigating whether a tCEC-based screening assay could distinguish between men with and without csPCa and assist in ruling out unnecessary biopsies.

## 2. Results

Patients were recruited for the study at Siriraj Hospital in Bangkok between July 2016 and October 2017, with a total of 170 patients enrolled. Three men were excluded due to inconclusive biopsy results (no prostatic acini obtained), three because of clotted blood samples, and 18 because of technical failure during tCEC analysis (Figure 1).

All blood was drawn from the cubital vein immediately before biopsy and processed locally in Bangkok within 48 h (Figure 2).

Of 146 men included in the study, 69 men showed PSA readings in the classic diagnostic grey zone below 10 ng/mL, 34 men had readings between 10 and 20 ng/mL, 23 men between 20 and 100 ng/mL, and 20 men over 100 ng/mL (Table 1).

Seventeen of 146 men with negative first TRUS biopsy received a second biopsy during follow-up. The median time between first and second TRUS biopsy was 9 months (2–20 months). As illustrated in Table 2, overall results of first and second TRUS biopsy showed that 24 (34%) of 71 patients with PSA <10 ng/mL, 15 (47%) of 32 patients with PSA 10–20 ng/mL, and 36 (84%) of 43 patients with PSA >20 ng/mL were positive for any cancer. Clinically significant cancer with Gleason score 3 + 4 or higher was found in 15 of 71 (21%) patients with PSA <10 ng/mL, 12 of 32 (38%) patients with PSA 10–20 ng/mL, and 36 of 43 (84%) patients with PSA >20 ng/mL, while csPCa with Gleason score 4 + 3 or higher was found in 9 of 71 (13%) patients with PSA <10 ng/mL, 8 of 32 (25%) patients with PSA 10–20 ng/mL, and 31 of 43 (72%) patients with PSA >20 ng/mL. Definition of clinically significant and clinically non-significant prostate cancer cases was based on the patients’ clinicopathologic factors, following the joint guidelines of the European Association of Urology, the European Society for Radiotherapy & Oncology and the International Society of Geriatric Oncology (EAU-ESTRO-SIOG Guidelines) as further detailed in Patients and Methods [25].

We then compared the outcome of the tCEC test to the results of the TRUS biopsy and computed diagnostic accuracies, as shown in Figure 3. In the classic diagnostic grey zone (PSA < 10 ng/mL), sensitivity of the tCEC assay for csPCa according to the primary definition was 75% (95% confidence interval (CI) 43–95), with a specificity of 67% (53–79). The PPV was 32% (16–52) with an NPV of 93% (80–98). According to the secondary definition, the sensitivity was 71% (29–96) with a specificity of 63% (50–75). The PPV was 18% (6–37), with an NPV of 95% (83–99).

Following the primary definition, only three of 41 men who had a negative tCEC test showed csPCa on TRUS biopsy. One of these had Gleason score 4 + 5/4 + 3 (left/right prostate) and T2cNxMx on clinical staging, the second had Gleason score 4 + 4/4 + 4 and T3aN0M0, and the third showed Gleason score 3 + 4 and T2bN0M0.

In two other men, the tCEC test predicted the presence of cancer in spite of a negative first TRUS biopsy. One man had a positive tCEC test, a first negative biopsy, and a second positive biopsy two months later that revealed Gleason score 4 + 3. He was then lost to another hospital for prostatectomy. A second man positive for tCEC but negative on first TRUS biopsy received an mpMRI scan 17 months later that revealed bladder cancer without any visible lesion in the prostate. Results for patients with PSA outside the diagnostic grey zone (PSA > 10 ng/mL) are shown in Appendix A.

Among the 17 men who received a second TRUS biopsy after a first negative result, five cases of PCa were detected. Of these, a Gleason score 3 + 4 or higher was found in four and Gleason score 4 + 3 or higher in three. All had been cases with positive tCEC test. On the other hand, all second TRUS biopsies in men with originally negative tCEC tests were PCa negative (Appendix A).

With NPVs over 90% for men with PSA values in the diagnostic grey zone for both the primary and the secondary definition of csPCa, we next compared the PPV of the tCEC test with the PPV of PSA alone. The difference between the PPVs of PSA and tCEC test was significant for the primary endpoint (all PCa) (32% vs. 54%, general estimating equation (GEE) model odds ratio 2.47 (95% CI 1.43–4.24); *p* = 0.0011), as well as the secondary endpoint, primary definition (Gleason 3 + 4) (17% vs. 32%, GEE model odds ratio 2.25 (95% CI 1.38–3.68); *p* = 0.0012) and secondary definition (Gleason 4 + 3) (10% vs. 18%, GEE model odds ratio 1.93 (95% CI 1.09–3.40); *p* = 0.0012). 

In contrast, alongside smaller cancer-positive patient numbers for men above the diagnostic grey zone, differences between the PPV of the tCEC test and PSA were not significant (all *p* > 0.05) (Table 3). In general, for patients with PSA values above the diagnostic grey zone, the diagnostic benefit seemed less pronounced: the NPV of the tCEC test decreased while the PPV of PSA increased (Appendix A).

Based on these findings, we estimated the clinical utility of using tCEC testing as an add-on triage test in patients with elevated PSA, assuming that only men with a positive tCEC test would go on to biopsy (Appendix A).

Under such a scenario—both following the primary and secondary definition—41 (59%) of 69 of total primary biopsies (95% CI 47–71), or 34 (72%) of 47 negative primary biopsies (57–84) in the diagnostic grey zone would have been avoided. In turn, this could lead to a relative reduction of four (40%) in 10 men (12–74) and five (33%) in 15 (12–62), respectively—avoiding over-diagnosis of clinically insignificant cancer for both primary and secondary definition. The complete calculation for this scenario including patients above the diagnostic grey zone is shown in Appendix A.

## 3. Discussion

To our knowledge, this is the first study to present prospective, blinded data on the diagnostic accuracy of tCEC in biopsy-naïve men both within and above the diagnostic grey zone. While relatively small in patient numbers and single-centre, it represents the first level-2 evidence for this biomarker in prostate cancer [26].

The most prominent outcome is the NPV exceeding 90% for both the primary and the secondary definition of csPCa. Given the fact that—depending on the clinical definition of csPCa—the PPV of PSA alone was found to be as low as 10% for csPCa, we calculated that a tCEC-based screening tool could safely avoid more than 70% of all negative prostate biopsies on patients in the diagnostic grey zone if used as a rule-out test between PSA and biopsy (Appendix A). Equally noteworthy is that over-diagnosis of clinically insignificant cancers may be reduced by up to 40%.

Within the limited data available in this study, tCEC testing identified nearly as many false-negative TRUS biopsies as it missed csPCa. Building on previous findings demonstrating the significance of tCEC in PCa as well as in other cancer types and—for the first time—based on clearly defined cytopathological criteria (Table 4), the results present further progress towards clinical routine use of tCEC [14,27].

Intriguingly, the number of tCEC cells observed did not correlate to clinicopathologic factors—in other words, to the risk classification of patients. Given the limited size of the current study, however, this observation will need confirmation in larger follow-up studies. Also, classical, epithelial CTCs were observed in only four patients diagnosed positive for PCa. This detection rate seems to be lower than reported from previous CTC detection studies. However, most CTC studies have focussed on follow-up of late-stage PCa patients and are therefore difficult to compare. Another difference may originate from the fact that CTC studies usually rely on an antibody panel detecting cytokeratins 8, 18, and 19, while the present study employed an anti-pan-cytokeratin antibody targeting cytokeratins 4, 5, 6, 8, 10, 13, and 18, but not 19. Further studies will need to address and examine this observation in greater detail [25,28].

A number of limitations need to be considered. With a false-negative rate of over 30%, TRUS biopsy is not an ideal reference test. Multi-parametric MRI (mpMRI) for detection of csPCa was not routinely available at the clinical site when this study was initiated but has since been adopted in most urology guidelines and gained wide-spread acceptance amongst clinicians. However, with recent studies still reporting a 24% false-negative rate for mpMRI, results of the present study may only have marginally benefited from this technology [13,29]. Nonetheless, with mpMRI now integrated into clinical routine in many centres, follow-up studies putting both technologies into perspective are certainly warranted.

The present study explored the utility of tCEC within and above the diagnostic grey zone. The number of patients in the diagnostic grey zone below 10 ng/mL was 69, with an additional 34 with PSA between 10 and 20 ng/mL. While this represents a relatively small cohort, most results for patients in the diagnostic grey zone proved statistically significant, especially the reduction of biopsies while maintaining an NPV above 90%.

Results from this relatively small single-centre study cannot be readily extrapolated to other geographical settings. It should be noted, however, that the prevalence of cancer observed in our study population is reflected well in other regional studies, and that so far cytopathology itself has been a universal, ethnically unbiased diagnostic approach (Figure 1) [30,31].

Patients with negative PSA readings (<4 ng/mL) were excluded from our study. While this is in line with current clinical practice at the study site, it precludes calculation of sensitivity, specificity, and NPV of PSA alone and hence any comparison with the values obtained from the tCEC test.

Our single-laboratory setting did not allow any determination of inter-laboratory variations, such as inter-operator variability during both the wet lab part and microscopic analysis of slides. The relatively high failure rate with 18 patients lost due to technical issues can be traced back to the fact that inter-operator validations needed improvement during the early days of the study. Multi-site assay validation has since been initiated to avoid similar issues during follow-up studies.

Following best practice in set-up for diagnostic accuracy studies and in line with recent prostate cancer screening landmark studies, such as the PRECISION and PROMIS trials, the study adopted a single-gate, prospectively blinded design, not including any healthy controls. Adding healthy controls as a second gate to diagnostic accuracy studies is well described to result in “inflated estimates of diagnostic accuracy”; however, research into circulating rare cells in healthy donors in separate studies may yield valuable additional insights on cells related to other (prostate) conditions, and may help to explain the false-positives observed in the current study [13,32,33].

Clinical evidence obtained in the present study clearly demonstrates a statistically significant association of tCECs with the presence of prostate cancer. However, in the absence of prostate-specific markers on endothelial cells, it may be hypothesised that cells originate from endothelium elsewhere in the body, potentially triggered systemically by the presence of prostate cancer. While it has already been shown in colon cancer that the latter scenario is not the case—colon cancer-associated circulating endothelial cells were found to originate from the primary tumour—further research is warranted to answer this question for prostate cancer [14].

Finally, the current study focused exclusively on the biopsy-naïve patient, excluding patients with known prostate cancer. The utility of tCEC detection as a therapeutic adjunct or for relapse monitoring, however, may be a worthwhile consideration.

## 4. Patients and Methods

In the present single-centre, prospective, blinded, paired-cohort study, we included 170 men with clinical suspicion of PCa and scheduled for prostate biopsy. Inclusion criteria were elevated serum PSA (equal to or higher than 4 ng/mL) within the previous three months, suspicious digital rectal examination, as well as critical age or family history. Patients were excluded if they were not able to give informed consent, had already undergone biopsy within the past three months, or if they had another known malignancy.

Patients were enrolled at the Urology Department of Siriraj Hospital, Bangkok, Thailand between July 2016 and October 2017 in convenience series. Procedures were approved by the hospital’s internal review board, certificate of approval no. 402/2015, and complied with the Declaration of Helsinki. All human participants gave written informed consent.

The blood required for tCEC analysis was drawn from the cubital vein immediately before a scheduled TRUS prostate biopsy. All samples were processed within 48 h at our laboratory at the Faculty of Science, Mahidol University, Bangkok, Thailand. Both equipment and reagents for high-gradient magnetic cell separation (hMX) and cryo-immunostaining were provided by X-ZELL, Singapore.

Each sample underwent red blood cell lysis and CD45-based high-flow magnetic white blood cell depletion. All materials and reagents employed for the following protocols are listed in Appendix A.

Twenty millilitres of hMX lysis buffer were briefly added to 5 mL of whole blood before incubating the samples for 5–7 min (until clear) and washing them with phosphate buffered saline containing ethylenediaminetetraacetic acid 5 mM and fetal bovine serum 1% (PBS/EDTA 5 mM/FBS 1%) at 400× *g* for 10 min at the lowest centrifuge acceleration and deceleration at room temperature. The cells were then re-suspended in 50 µL PBS/EDTA 5 mM/FBS 1% and counted in a haemocytometer.

Next, cells were blocked with Fc-receptor blocking reagent for 15 min at 4 °C before adding 1 μL each of biotin-anti-CD45 antibody and biotin-anti-CD235 antibody and incubating them for 15 min at 4 °C. They were then washed once in PBS/EDTA 5 mM/FBS 1% at 400× *g* for 10 min, with the supernatant being discarded. One hundred microlitres of hMX anti-biotin nanobeads were added before the cells were incubated for another 15 min. Finally, 1 mL of hMX buffer was added and separation performed on gravity-fed hMX separation columns mounted in an hMX separator following the manufacturer’s protocol. In summary, the cell suspension was applied to the inlet of the separation column, the stopcock opened, and the column washed with 20 mL hMX buffer. The flow-through was collected, centrifuged at 1000× *g* for 10 min, and the supernatant discarded. Cells recovered in the flow-through were counted in a haemocytometer. All incubations were performed on ice.

The remaining cells were subjected to multiplexed cryo-immunostaining with antibodies directed against CD31, CD34, CD45, Vimentin, pan-Cytokeratin, and EpCAM. Cryo-immunostaining was chosen as a cost- and time-efficient alternative to flow cytometry, as it is capable of applying up to nine antibodies to a slide-based sample at the same time without the need for downstream image processing, as described previously [23,24].

Cells were re-suspended in 700 µL Cytocentrifugation buffer and spun onto gelatine-coated slides in one-well concentrators of a StatSpin Cytofuge 2 (Beckman Coulter, Atlanta, GA, USA) for 10 min at 600 rpm at room temperature. Next, slides were fixed in Cryofixation Buffer I for 15 min at −25 °C and re-hydrated in Cryofixation Buffer II for 20 min at −2.5 °C in a Cryofixation Station. Subsequently, slides were mounted on CapGap clips. CapGap-slide assemblies were mounted in the Cryostainer, which provided a staining temperature of −2 °C. Two microlitres of Fc-blocking reagent in 100 µL blocking buffer were applied, followed by cocktails of primary-conjugated antibodies in antibody binding buffer. Incubation times were 45 and 60 min for blocking and staining, respectively. After each step, slides were washed with 200 µL antibody binding buffer. One hundred microlitres of pre-mounting buffer were applied, slides unmounted and coverslipped with 25 µL MB I buffer containing DRAQ5 DNA dye.

Finally, slides were analysed on a fully motorised DM6000B fluorescence microscope (Leica, Wetzlar, Germany) equipped with excitation and emission filter wheels (Sutter, Atlanta, GA, USA), a 9-colour zero-crosstalk fluorescence filter set (X-ZELL, Singapore, Singapore), and an Orca ER monochrome digital CCD camera (Hamamatsu, Tokyo, Japan). The microscope assembly was controlled from a computer running MicroManager 1.4 software (Open Imaging, San Francisco, CA, USA).

Atypical cells were classified according to pre-defined immunocytopathological criteria derived from the literature and own data (see Table 4) [14,27,34,35]. All cases with cell class III (suspicious for malignancy) or higher were considered positive for malignancy.

Following the blood draw, 10–12 core TRUS prostate tissue biopsies were performed following international standards. In some cases, a second TRUS biopsy was carried out to verify inconclusive findings of the first biopsy. Researchers and pathologists were blinded towards the tCEC test result as well as tissue biopsy results, respectively.

For both tCEC assay and reference test, the primary endpoint was the presence of any kind of prostate cancer on TRUS biopsy, while the secondary endpoint was the presence of csPCa only.

Following the EAU-ESTRO-SIOG Guidelines 2016 and previous studies in the field, the clinicopathological parameters of PSA, Gleason score, and T-stage were condensed into low-, intermediate-, and high-risk groups. Clinically significant PCa in the primary definition was defined as any high-risk cancer and as intermediate-risk cancers with Gleason ≥3 + 4 or higher (Grade Group 2 or higher). Clinically significant PCa in the secondary definition was defined as any high-risk cancer and as intermediate-risk cancers with Gleason ≥4 + 3 (Grade Group 3 or higher) as a secondary definition [13,25,29,33] (Table 5).

Results of diagnostic accuracy for the tCEC test are presented in 2 × 2 contingency tables with 95% confidence intervals. In addition, the assay’s performance is compared to the performance of PSA at pre-defined cut-offs of 10 and 20 ng/mL, which reflect both the classic as well as the extended diagnostic grey zone as defined in a range of recent studies [10,36].

For comparison of the PPV of the tCEC test with the PPV of PSA, a GEE logistic regression model was employed [37,38,39]. Analysis was performed using SAS Studio 9.

## 5. Conclusions

Recent high-profile studies have demonstrated no cancer-specific survival benefit when choosing active monitoring over radical prostatectomy or radical radiotherapy, although there may be a beneficial effect in reducing disease progression and metastases [4]. These findings emphasise the need for improved diagnostic rule-out tests for clinically insignificant disease and healthy patients, especially with mpMRI unable to fill the gap. Despite demonstrating reasonably high PPV for csPCa across multiple studies, NPVs reported for mpMRI fluctuate significantly depending on study set-up and the definition of csPCa. Given the results of the present study, a tCEC-based assay—if used as a PSA add-on—may therefore serve as a rule-out test that, if positive, could be followed by mpMRI as a rule-in test.

To sum, this study presents the first evidence that tCECs may serve as a novel liquid biopsy, potentially avoiding over half of all prostate biopsies and significantly reducing the risk of over-diagnosing clinically insignificant prostate cancer.

## Figures and Tables

**Figure 1 cancers-11-01064-f001:**
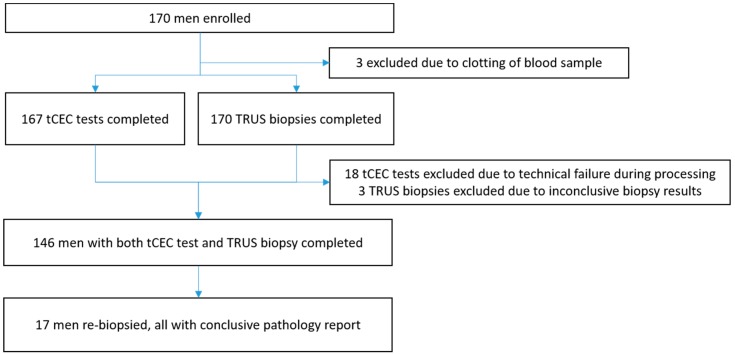
Trial flow diagram. tCEC: tumour-associated circulating endothelial cells; TRUS biopsy: transrectal ultrasound-guided biopsy.

**Figure 2 cancers-11-01064-f002:**
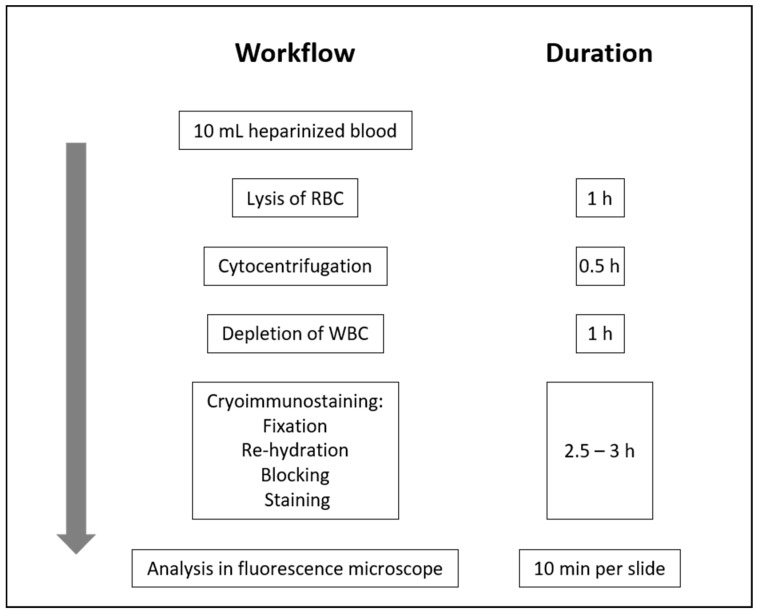
Workflow of tumour-associated circulating endothelial cell detection. WBC: white blood cells; RBC: red blood cells.

**Figure 3 cancers-11-01064-f003:**
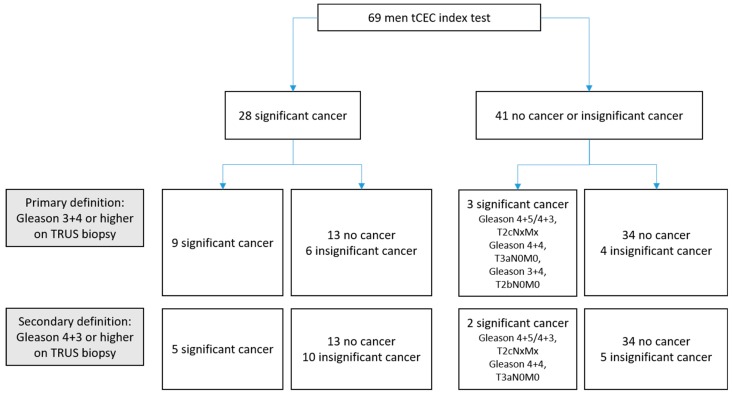
Diagnostic accuracy of the tCEC test for detection of clinically significant cancer in men with PSA readings <10 ng/mL on TRUS biopsy. tCEC: tumour-associated circulating endothelial cells, TRUS biopsy: transrectal ultrasound-guided biopsy. Primary definition: Sensitivity 75% (95% confidence interval (CI) 43–95), specificity 67% (53–79), positive predictive value 32% (16–52), negative predictive value 93% (80–98). Secondary definition: Sensitivity 71% (29–96), specificity 63% (50–75), positive predictive value 18% (6–37), negative predictive value 95% (83–99).

**Table 1 cancers-11-01064-t001:** Baseline characteristics of 170 men enrolled. PSA: prostate-specific antigen. NA: not applicable.

PSA Level	146 Men Included					24 Men Excluded				
	Men, *n*	Mean Age	SD	Mean PSA	SD	Men, *n*	Mean Age	SD	Mean PSA	SD
<10	69	61.4	20.1	6.5	2.2	12	69.9	8.6	5.9	2.2
10–20	34	66.2	13.4	14.0	2.7	7	64.4	3.0	12.1	6.5
20–40	13	64.5	21.1	30.3	7.3	3	68.7	3.2	28.2	6.2
40–60	5	65.8	9.8	51.0	7.0	0	NA	NA	NA	NA
60–80	5	69.8	7.5	69.8	4.4	1	76.0	NA	60.7	NA
80–100	0	NA	NA	NA	NA	0	NA	NA	NA	NA
>100	20	69.9	8.6	1271.9	1188.8	1	73.0	NA	1114.0	NA
Total	146					24				

**Table 2 cancers-11-01064-t002:** Number of men with no cancer and clinically significant prostate cancer (primary and secondary definition) on TRUS biopsy. Clinically significant prostate cancer was defined based on the patients’ clinicopathologic factors as described in patients and methods. PCa: prostate cancer; csPCa: clinically significant prostate cancer; PSA: prostate-specific antigen; TRUS biopsy: transrectal ultrasound-guided biopsy.

PSA ng/mL	All PCa			csPCa, Gleason ≥ 3 + 4			csPCa, Gleason ≥ 4 + 3		
	TRUS Biopsy+	TRUS Biopsy−	%	TRUS Biopsy+	TRUS Biopsy−	%	TRUS Biopsy+	TRUS Biopsy−	%
**<10**	24	47	34	15	56	21	9	62	13
**10–20**	15	17	47	12	20	38	8	24	25
**20–40**	7	6	54	7	6	54	4	9	31
**40–60**	4	1	80	4	1	80	4	1	80
**60–80**	5	0	100	5	0	100	3	2	60
**80–100**	0	0		0	0		0	0	
**>100**	20	0	100	20	0	100	20	0	100
**Total**	**75**	**71**		**63**	**83**		**48**	**98**	

**Table 3 cancers-11-01064-t003:** Comparison of PPV of PSA and tCEC test for all prostate cancer and clinically significant cancer in men with PSA readings <10 ng/mL (A) and 10–20 ng/mL (B) according to the primary and secondary definitions. PPV: positive predictive value; PSA: prostate-specific antigen; CI: confidence interval; tCEC: tumour-associated circulating endothelial cells

**A**	**PSA < 10 ng/mL**				
		**PSA, % (95% CI)**	**tCEC, % (95% CI)**	**Test Ratio * (95% CI)**	***p* Value ***
	Primary endpoint, prevalence of all cancer: *n* = 22				
	**PPV**	32 (21–44)	54 (34–73)	2.47 (1.43, 4.24)	0.0011
	Secondary endpoint, primary definition, prevalence of clinically significant cancer with Gleason score 3 + 4 or higher: *n* = 12				
	**PPV**	17 (9–28)	32 (16–52)	2.25 (1.38, 3.68)	0.0012
	Secondary endpoint, secondary definition, prevalence of clinically significant cancer with Gleason score 4 + 3 or higher: *n* = 7				
	**PPV**	10 (4–20)	18 (6–37)	1.93 (1.09, 3.40)	0.0242
**B**	**PSA 10–20 ng/mL**				
		**PSA, % (95% CI)**	**tCEC, % (95% CI)**	**Test Ratio * (95% CI)**	***p* Value**
	Primary endpoint, prevalence of all cancer: *n* = 16				
	**PPV**	47 (30–65)	60 (32–84)	1.69 (0.77, 3.69)	0.1904
	Secondary endpoint, primary definition, prevalence of clinically significant cancer with Gleason score 3 + 4 or higher: *n* = 12				
	**PPV**	35 (20–53)	46 (21–73)	1.60 (0.77, 3.33)	0.2050
	Secondary endpoint, secondary definition, prevalence of clinically significant cancer with Gleason score 4 + 3 or higher: *n* = 8				
	**PPV**	24 (11–42)	28 (9–56)	1.18 (0.52, 2.70)	0.6917

* General estimating equation (GEE) logistic regression model used to compare PPV. Ratios are presented as tCEC results relative to PSA results.

**Table 4 cancers-11-01064-t004:** Circulating endothelial cell (CEC)/ circulating tumour cell (CTC) cytopathological criteria. + positive; − negative; = positive or negative.

Class	Cytopathological Diagnosis	Immunocytomorphology
I	Negative for malignancy	No CD45− cells
II	Atypical cells but negative for malignancy	CD45− cells without positive markers (consider plasma cells) CD31+ CD34− VIM− CK− CD45− cells with aneuploidy (consider normal endothelial cells, see Lin et al. [27]) CD45+ cells with atypical nuclei
III	Suspicious for malignancy	Less than five single cells: CD31= CD34+ VIM+ CK− CD45− (angiogenic tip cell: tumour-associated vs. inflammatory) More than one large cell: CD31− CD34= VIM+ CK− CD45− (mesenchymal CTC vs. haematopoietic stem cell) Both conditions with or without aneuploidy
IV	Strongly suggestive for malignancy	Less than five single cells: CD31− CD34− VIM= CK+ CD45− (epithelial CTC or cell in epithelial-mesenchymal transition) One clump: CD31= CD34+ VIM= CK= CD45− More than one cell with aneuploidy: CD31+ CD34+ VIM+ CK= CD45− More than five single cells without aneuploidy: CD31+ CD34+ VIM+ CK= CD45− One or more large cell with aneuploidy: CD45−
V	Conclusive for malignancy	More than one clump: CD31= CD34+ VIM+ CK= CD45− More than five cells: CD31− CD34− VIM= CK+ CD45− (epithelial CTC or cell in epithelial-mesenchymal transition) One or more CD45− cell in atypical mitosis (chromosome missegregation) One or more giant polyploidic cells, CD45−

**Table 5 cancers-11-01064-t005:** Risk groups for biochemical recurrence of localised and locally advanced prostate cancer, as introduced in the joint guidelines of the European Association of Urology, the European Society for Radiotherapy & Oncology and the International Society of Geriatric Oncology 2016 [25].

Low-Risk	Intermediate-Risk	High-Risk	
Definition			
PSA <10 ng/mL	PSA 10–20 ng/mL	PSA > 20 ng/mL	any PSA
and GS <7	or GS 7	or GS >7	any GS
and cT1-2a	or cT2b	or cT2c	cT3–4 or cN+
Localised	Localised	Localised	Locally advanced

GS: Gleason score; PSA: prostate-specific antigen.

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
