# Peer review of "Accuracy of Tumour-Associated Circulating Endothelial Cells as a Screening Biomarker for Clinically Significant Prostate Cancer"

_cancers, 2019, doi:10.3390/cancers11081064_

Round 1
Reviewer 1 Report
This manuscript demonstrated that the accuracy enhancement for clinical prostate cancer diagnosis using tumour-derived circulating endothelial cells as a screening biomarker. It's interesting and helpful to improve the clinical diagnostic accuracy of prostate cancer using PSA as a biomarker. Some problems in this manuscript should be revised before publication in Cancers.
In the Introduction, the authors should provide an additional remark about the relationship between tumour-derived circulating endothelial cells and circulating tumor cells.
The authors should analyze the clinicopathologic factors associated with detection rate and count of tumour-derived circulating endothelial cells, circulating tumor cells, and PSA.
The authors should also do the comparison of PSA and tumour-derived circulating endothelial cells using biopsies from healthy people.
The authors should stain the tumour-derived circulating endothelial cells to prove the tumour-derived circulating endothelial cells are derived from prostate tumor indeed.
The authors should study the comparison of PSA and tumour-derived circulating endothelial cells using the biopsies before and after clinical treatment from a patient.
Reviewer 2 Report
The aim of the manuscript is to evaluate the accuracy of tCEC as screening Biomarker for csPCa. The manuscript is interesting and the topic is actual even if there are some critical aspects that Author has to review to improve the overall quality of the manuscript.
Author has reported that 18 patients were excluded due to technical failure during tCEC. It means that in about 10% of patients it was not possible the analysis of tCEC, the Author should describe in detail the reasons for which it was not possible the analysis.
Why it has not used the mpMRI for the detection of csPCa? It could be interesting evaluate the results after a mpMRI and a fusion biopsy to improve sensitivity and specificity of tCEC.
Round 2
Reviewer 1 Report
This manuscript can be accepted for publication in Cancers in present version.